# Hydroxylated Fatty Acids: The Role of the Sphingomyelin Synthase and the Origin of Selectivity

**DOI:** 10.3390/membranes11100787

**Published:** 2021-10-16

**Authors:** Lucia Sessa, Anna Maria Nardiello, Jacopo Santoro, Simona Concilio, Stefano Piotto

**Affiliations:** Department of Pharmacy, University of Salerno, Via Giovanni Paolo II, 84084 Fisciano, SA, Italy; annardiello@unisa.it (A.M.N.); jsantoro@unisa.it (J.S.); sconcilio@unisa.it (S.C.)

**Keywords:** SMS, 2—hydroxy oleic acid, metadynamics

## Abstract

Sphingolipids are a class of lipids acting as key modulators of many physiological and pathophysiological processes. Hydroxylation patterns have a major influence on the biophysical properties of sphingolipids. In this work, we have studied the mechanism of action of hydroxylated lipids in sphingomyelin synthase (SMS). The structures of the two human isoforms, SMS1 and SMS2, have been generated through neural network supported homology. Furthermore, we have elucidated the reaction mechanism that allows SMS to recover the choline head from a phosphocholine (PC) and transfer it to ceramide, and we have clarified the role of the hydroxyl group in the interaction with the enzyme. Finally, the effect of partial inhibition of SMS on the levels of PC and sphingomyelin was calculated for different rate constants solving ordinary differential equation systems.

## 1. Introduction

Sphingolipids are key molecules in regulating the cell cycle, apoptosis, angiogenesis, stress, and inflammatory responses. Sphingomyelin (SM) is an important structural component of biological membranes and one of the endpoints of sphingolipid synthesis. With phosphatidylcholine (PC), SM is one of the most abundant phospholipids in biological membranes. It is found in high concentrations in the outer leaflet of the plasma membrane, where it plays important structural roles. Sphingomyelin is synthesized primarily in the Golgi apparatus and then transported to all other biological membranes. It is produced by sphingomyelin synthase (SMS) in Golgi [1] and hydrolyzed to ceramide by five different sphingomyelinases. The structural diversity and the cellular topology allow ceramide to exert multiple effects and be metabolized into other bioactive sphingolipids. Some diseases that involve the sphingomyelin cycle include cancer, inflammation, atherosclerosis, diabetes, and some rare diseases [2,3,4]. The type and composition of sphingolipids modulate the biophysical properties of membranes [5,6,7], which can be organized into two-dimensional domains. Membrane properties determined by the specific type and abundance of sphingolipids allow biological membranes to adapt to temperature, pH, and membrane tension changes [8,9]. For example, the presence of SM increases the stiffness and compactness of the plasma membrane (PM) [10,11]. In mammalian membranes, the SMs with different acyl chains, together with unsaturated phospholipids and cholesterol, can be used by the cell to refine the lateral structure of the membranes [12,13].

Several authors have suggested that changes in membrane properties promoted by the composition of sphingolipids can trigger cell signaling [14]. However, the link between physical properties and cell signaling is complicated because of the many components and the characteristic dynamics of membranes [15]. Sphingolipids are central in determining the physical state of membranes and turn out to be bioactive molecules [16].

SM can be hydrolyzed to ceramide by alkaline, acidic, or neutral sphingomyelinases in the plasma membrane and other cellular compartments because the total amount of SM is normally more than 10 times the amount of total ceramide in the cell [17], hydrolysis of a small percentage of SM results in large changes in ceramide levels. Ceramide generated by this pathway is further degraded into sphingosine. The sphingolipid pathway is highly branched and interconnected. The same enzyme activities are represented by different enzymes in different organelles, interconverting one bioactive species into others, but these bioactive lipids can also be transported from one compartment to another. Misregulation of a sphingolipid enzyme can lead to the accumulation or depletion of one or more sphingolipid species in a specific organelle [18]. Intracellular sphingolipid accumulation or altered cellular signaling may induce a pathological condition [19,20,21]. One of the isoforms of SMS, SMS2, is also found in the plasma membrane. Both forms can also catalyze the reverse reaction (SM to ceramide) [22].

SMSs are virtually present in all tissues, and SMS1 appears to be responsible for most of the SMS activity in most cells [23]. The two isoforms share 57% sequence identity and are conserved in mammals [23,24]. SMS1 contains a sterile alpha motif (SAM), involved in protein–protein interaction, which is not present in SMS2. SMSs are related to the lipid phosphatase (LPP) family, with six transmembrane regions with N- and C-terminals exposed in the cytosol [10]. Although SMS requires PC as the headgroup donor to form SM, overexpression or knockdown of SMS primarily affects sphingolipid levels (SM and ceramide) without notable changes in PC levels [25,26]. Although diacylglycerol (DAG) can be rapidly reincorporated into the PC, some studies suggest that SMS-derived DAG can trigger localized cellular responses, such as protein kinase D (PKD) translocation in the Golgi [25].

As SMS and sphingomyelinases are linked to multiple diseases, some authors predict [17] that more drugs targeting the SM cycle will be developed in the future.

An additional feature in all classes of sphingolipids shared with other ceramides related phospholipids is the possibility of being hydroxylated [27]. Sphingolipid hydroxylation, either in the acyl chain or the long-chain base (LCB), i.e., in the backbone of the sphingolipid [28,29,30] can also affect membrane lipid packing and regulation of G-protein [31]. Hydroxylation patterns have a major influence on the biophysical properties of sphingolipids, as illustrated, for example, by the significant difference in the disordered gel-liquid phase transition temperature (L_d_) when comparing similar sphingolipids with different hydroxylation patterns [32,33]. More important, perhaps, is the influence of hydroxylation in the interaction between sphingolipids and the surrounding membrane containing other lipid components, i.e., ester-bound glycerophospholipids and even sterols. Recently, important studies of membrane interaction with non-sphingolipid compounds containing OH groups have been reported [34].

Sphingolipids containing 2-hydroxylated fatty acids (2OHFA) are present in most organisms [32] and are important components of a subset of mammalian sphingolipids [35]. The enzyme FA2H (fatty acid 2-hydroxylase) is a hydroxylase that introduces a hydroxyl group into the 2-position of fatty acids [36].The 2-hydroxy fatty acids are found almost exclusively as N-acyl chains within the ceramide fraction of various sphingolipids [37]. FA2H is stereospecific to produce (R)-2-hydroxy fatty acids [38]. 2-Hydroxylation occurs during de novo synthesis of ceramide and is catalyzed by fatty acid 2-hydroxylase [35]. In mammals, all six isoforms of CerS (Ceramide synthases) can use 2-hydroxy acyl-CoA as substrates to synthesize 2OHFA-dihydroceramide [39]. In addition, galactosylceramide synthase has been shown to have a strong preference for 2OHFA ceramide over non-hydroxylated ceramide [40].

The influence of hydroxylation can be more investigated by comparing how hydroxylated, and non-hydroxylated lipids interact with related enzymes.

Unfortunately, not much data exists on SMS1 and SMS2 [41], and this work aims to address this paucity. Even less is known about how hydroxylated lipids may interact with SMS. The mechanism of action of the two isoforms of SMS will be illustrated, and the change in free energy in the intermediate stages will be estimated. Defining the intermediate stages allows clarification of the role of hydroxylation on the alpha carbon in Cer, PC, and SM chains.

## 2. Materials and Methods

### 2.1. Structure Prediction and Validation

SMS1 is formed by 413 residues. It can be organized into three parts: two cytosolic fragments (N- and C-terminal) and the transmembrane portion. SMS2 is formed by 365 residues, and the main difference with SMS1 is the lack of the SAM domain. The sequences in fasta format were downloaded from the Uniprot database (for human SMS1 code Q86VZ5, for human SMS2 code Q8NHU3). Sequence alignment of the two isoforms was provided by Clustal Omega multiple sequence alignment program [42]. The tertiary structure was predicted by the de novo structure prediction based on Folden modeling suite [43]. The algorithm uses a deep residual neural network to predict the inter-residue distance and orientation distributions of the input sequence. The models of the two isoforms of SMS with the best TM-score were validated by PROCHEK v.3.5 web server [44]. We used WHAT IF [45], a widely used program for structure validation, to compare the 3D structure of SMS1 built in this work with our previous work. For more details about the quality of the structures, see Appendix A.

### 2.2. Binding Site Definition

The conservation string was obtained by the Consurf database [46], a server for identifying structurally important residues in protein sequences. The conservation string ranges from 9 for very conserved residues to 1 for no conserved amino acids, as described in ref. [47].

The Waterscope tool [43] was set with a cuboid simulation cell of 5 Å around the receptor. The coordinates of each atom of the receptor were fixed, and the system was neutralized with NaCl at a concentration of 0.9%. The charges were assigned at pH 7.0 with the force field AMBER15IPQ [48]. Water molecules were modelled Tip3P with a density of 0.997 g/mL [49]. We have used the Berendsen thermostat at 298 K with the integration time steps for intramolecular forces every 1.25 fs. After the neutralization phase of the system, we have performed a 50 ns molecular dynamics simulation using the SolventProbe Yasara’s barostat. We have monitored the changes in H-bond patterns that water molecules, which are less than 4 Å away from the receptor, make throughout the dynamics at 50,000 fs intervals.

### 2.3. Molecular Docking

We used molecular docking to collect the geometry of the complex SMS1 and SMS2 with the natural substrates and the hydroxylated analogs. The simulations were performed using Autodock VINA [50] in the YASARA Structure package [51] and with the software Yada [47]. The use of these two software permits to reach a consensus both in pose geometry and energy calculation. The selected force field was AMBER14 [52] for both software. With Autodock VINA, the ligands were independently docked 250 times with 5 receptor ensembles with alternative high-scoring solutions of the side chain rotamer network. The simulation cell was defined around the key residue Tyrosine 223 for SMS1 and Tyr167 for SMS2. The results were clustered with a RMSD of 5.0 Å. With Yada, we used the same total blind docking procedure described in our previous works [53,54]. We have chosen 250 independent runs per hotspot (the barycenter of proximal conserved residues) in a box 20 Å larger than the receptor.

We have considered the natural substrates of the SMS enzyme, PC, SM, Cer and DAG, and their hydroxylated forms for a total of 12 ligands (Figure 1).

In detail, the ligands docked on both isoforms were: 1-palmitoyl-2-oleoyl-sn-glycero-3-phosphocholine (PC), hydroxylated PC with 2ROHOA (2ROHPC), hydroxylated PC with 2SOHOA (2SOHPC), *N*-oleoyl-d-erythro-sphingosylphosphorylcholine (SM), hydroxylated SM with 2SOHOA (2SOHSM), hydroxylated SM with 2ROHOA (2ROHSM). Additionally, we have considered the phosphorylated form of the protein indicated as SMS1-P and SMS2-P. We modified the tyrosine 223 in SMS1 by adding the phosphocholine group to the sidechain. We did the same for the tyrosine 167 in SMS2. We used these activated forms of the target to perform the docking with the natural substrates of SMS enzyme, such as DAG and Cer. In detail, the ligands docked on both isoforms were: (S)-1-hydroxy-3-(palmitoyloxy)propan-2-yl oleate (DAG), hydroxylated DAG with 2ROHOA (2ROHDAG), hydroxylated DAG with 2SOHOA (2SOHDAG), ceramide *N*-((2S,3R,E)-1,3-dihydroxyoctadec-4-en-2-yl)oleamide (Cer), hydroxylated Cer with 2ROHOA (2ROHCer), hydroxylated Cer with 2SOHOA (2SOHCer). To prevent damages to the initial model, all systems were minimized by running combined steepest descent and simulated annealing keeping the backbone atoms of the receptor fixed. The binding energy was calculated for the ligand object in YASARA with the AMBER14 force field [52].

We calculated the binding energy of the 12 ligands embedded in a PC membrane. PC lipid bilayer was used as the model. Each monolayer of the membrane consisted of 60 lipids. An initial periodic simulation cell (X = 80.25 Å, Y = 79.11 Å, Z = 90.15 Å) was built around the entire complex. The molecular dynamics simulations (MD) were performed using the software YASARA Structure 21.6.16 [51]. The charges were assigned at physiological conditions (pH 7.4). We used AMBER14 as a force field with long-ranged PME potential and a cutoff of 8.0 A. The simulation box was filled with Tip3P water, choosing a density of 0.997 g/mL. The system was neutralized with NaCl at a concentration of 0.9%. The membranes were equilibrated during 250 ps. The simulation was then initiated at 298 K and integration time steps for intramolecular forces every 1.25 fs. The simulation snapshots were saved at regular time intervals of 100 ps. The total simulation time was 100 ns.

The binding energy of the PC molecule was calculated as the average of the binding energy of 8 PC lipids dipped in the membrane formed by 120 PC residues. For hydroxylated PC with S configuration (2SOHPC) and R configuration (2ROHPC), 8 residues of a membrane of 120 PC lipids were replaced by the corresponding hydroxylated PC analogs. The same step was followed for DAG, Cer, and SM (and their corresponding hydroxylated analogs). All involved structures were minimized by running combined steepest descent and simulated annealing by fixing the backbone atoms of the aligned residues to avoid potential damage to the initial model. The MD simulation was performed for all new systems. The binding energy was calculated for each molecule as described above.

### 2.4. Molecular Dynamics Simulations

We analyzed the structural stability of the protein–ligand complexes through molecular dynamics simulations. The docked poses of protein–ligand complexes were used as input structures, and each complex was prepared by the system setup option in the Desmond module [55]. First, the protein–ligand complexes were pre-processed using the Protein Preparation Wizard in Maestro 2021-1 suite obtained through Desmond academic license. The missing hydrogens were added, bond orders were assigned, and the protein was minimized using the OPLS3e force field [56]. Next, the simulation system was prepared using the system builder wizard. The systems were centered in an orthorhombic box with the edges 10 Å away from the protein in all directions. The solvated model of a complex was prepared by selecting PC (300 K) as a membrane model. The orientation of SMS1 and SMS2 in membranes was predicted by using the OPM web server [57]. The tilt angle was very similar between the two proteins (23 ± 1° for SMS1 and 25 ± 1° for SMS2). The system was solvated in an orthorhombic box (Tip3P model) and neutralized with Na^+^ and Cl^−^ ions. The salt concentration was set as 0.15 M to maintain physiological conditions. The MD simulations were conducted with the periodic boundary conditions in the NPT ensemble using the OPLS3 force field. The temperature and pressure were kept at 300 K and 1013 bar, respectively, using Langevin temperature coupling and isotropic scaling. The operation was followed by a 30 ns NPT production run. The MD simulations were analyzed to monitor the ligand atom interactions with the protein residues and the protein interactions with the ligand throughout the simulation.

### 2.5. Metadynamics Simulations

We used the phosphorylated form of SMS (SMS1-P and SMS2-P) embedded in a POPC membrane for metadynamics simulations. We employed GPU accelerated Desmond software on an NVIDIA GeForce GTX 980 graphic card, using Langevin chain thermostat and barostat. A combination of two collective variables (CVs) that describes the ceramide (and hydroxylated analogues) movement in the binding site of the phosphorylated protein was defined. For the distance CVs, the Gaussian width was set to 0.05 Å. The starting height of the Gaussian potential was set to 0.03 kcal/mol, and the Gaussians were deposited every 0.09 ps. The simulations were performed at 300 K and 1.013 bar pressure. RESPA integrator was used with a time step of 2.0 fs. For coulombic interactions, short-range cutoff radius was defined at 9 Å. No positional restraints were specified for any of the atoms. The trajectory frames were recorded at an interval of 20 ps for a simulation time of 30 ns. The simulations were visualized in Maestro suite [4]. Analysis of the trajectory was performed using Simulation Event Analysis of Maestro and Visual Molecular Dynamics.

### 2.6. Reconstruction of the SMS Pathway

This study reconstructed a simplified SM synthesis network by integrating the sphingomyelin biosynthesis pathway, as shown in Appendix A. Interactions between components were represented as elemental chemical reactions in the SimBiology toolbox of MATLAB (2021a) [58] using the Systems Biology Markup Language (SBML) machine language.

Complex biological systems such as the sphingolipids network can be viewed as a system of chemical reactions that can be analyzed mathematically using ordinary differential equations (ODEs), which is the most common simulation approach used in computational systems biology. ODEs help determine time-dependent changes, i.e., time series data of signaling protein concentrations and protein complexes and thus associated dynamics [59]. The network was numerically simulated using the Stiff Deterministic ODE15s solver (SimBiology toolbox) that generates first-order nonlinear ordinary differential equations for each node, thus defining the mathematical structure of the model. Then, the model was exported as an SBML file to generate the time series data.

## 3. Results and Discussion

### 3.1. Structure Prediction and Validation: The Two Isoforms Show a High Homology Sequence

The two main isoforms of SMS, SMS1 and SMS2, share important sequence homology except for the SAM (sterile alpha motif) domain, a cytosolic domain present only in isoform 1. SMS1 consists of a sequence of 413 residues, whereas SMS2 consists of 365 amino acids. The TM portion (SMS1 131–353—SMS2 74–294) has a percentage sequence similarity of 74.55%.

In our previous work [24], we built the structure of SMS1 by homology modeling. However, improvements in deep learning-based folding algorithms have obtained much more accurate three-dimensional structures in recent years due to improvements in deep learning-based folding algorithms. The tertiary structures of the two isoforms of SMS were predicted using the Folden software in the SMP modeling suite [43]. The 3D models were subjected to the PROCHECK server, where Ramachandran plot statistics were generated.

For SMS1, the output showed 93.9% residues in the most favored region, 5.2% residues in the additional allowed region, 0.3% residues in the generously allowed regions and 0.6% residues in disallowed regions (Appendix A). For SMS2, the output showed 90.3% residues were present in the most favored region, 8.1% residues in the additional allowed region, 0.6% residues in the generously allowed regions and 0.9% residues in disallowed regions (Appendix A).

The two SMS isoforms showed a sequence homology percentage of 62.07% over the entire structure (Appendix A), but the cytosolic portions are less conserved than the transmembrane portion. Therefore, we compared only the transmembrane portion. In agreement with the Phobius ref prediction, for SMS1 we selected 222 residues from residue E131 to residue Q353, and from residue E75 to residue E297 was selected for SMS2 (see Appendix A). In detail, the two transmembrane portions shared 91.9% of fully conserved residues and residues with strongly similar properties (Appendix A). Using docking and molecular dynamics studies, we have tried to define whether these slight differences could affect the type of interactions in the binding site.

### 3.2. Binding Site Identification: The Transmembrane Portion Contains the Binding Site

We hypothesized that the most conserved portions in the two proteins played a crucial role in the enzyme’s activity. We used Consurf [46] to estimate the evolutionary conservation of amino acid positions in a protein-based phylogenetic relationship between homologous sequences [60]. The magenta-colored regions in Figure 2a,b are the most conserved regions between the two proteins and indicate the highest probability for the binding site.

It is possible to identify more accurately an active site by monitoring the movement of water molecules during molecular dynamics. For a ligand to interact with a receptor, the water molecules present in the binding site must be moved away. A dedicated tool, Waterscope [43], calculates several physicochemical parameters of water molecules, such as their diffusivity and intrinsic entropy. The residues at the water molecules that will have a greater entropic benefit in the bulk passage are those most likely to reside at the binding site.

As shown in Figure 2c, the core of the transmembrane portion corresponds to the location of low mobility water molecules. Stationary molecules in the transmembrane localize the catalytic site in this portion. Remarkably, the portion highlighted with a blue surface (Figure 2c) matches the highly conserved transmembrane portion displayed in Figure 2a,b. The SAM domain is also a catalytic domain, but SMS mutants with deletion of the SAM domain from SMS1 showed no significant impact on SMS catalytic activity [61]. Therefore, we concentrated on the transmembrane portion. The enzyme is immersed in a reservoir of lipids that can be processed. The natural substrates of SMS, PC and SM are poorly soluble molecules in water and can only access the membrane-immersed portions of the enzyme. The membrane components can easily reach the binding site in the transmembrane portion by moving freely through the membrane.

### 3.3. Study of the Mechanism of the Reaction: Tyrosine Is a Key Residue

Molecular docking allows a rough estimation of binding energies and definition of ligand interactions with receptors. We used software Autodock VINA [50] and Yada [47] to get a consensus assessment. Autodock VINA represents one of the most popular software for molecular docking. Yada is a docking software that combines structural and phylogenetics data to perform the investigation.

Analyses were conducted on both isoforms. The best docked pose of each complex was used as the input structure for molecular dynamics (MD) simulations. The two isoforms were embedded in a POPC membrane.

Molecular dynamics trajectory analysis for isoform 1 revealed a contact for 99% of the time between Tyr223 and the phosphate group of the substrate. This permits a nucleophilic attack of the hydroxyl group of tyrosine 223 to the phosphorus atom of the PC molecule. Thus, the DAG molecule can move away from the active site after O-P bond formation and diffuse into the membrane (Figure 3).

In addition, the presence of a water molecule completes a cyclic network by hydrogen bonding to the Tyr223 and the phosphate group of PC (see also Figure 4), and can assist the nucleophilic substitution on phosphatidylcholine.

Trajectory analyses indicate the role of two other residues, the Tyr280 and His 285, in anchoring the phosphocholine head to the target (Figure 4a). Tyr167 plays a similar role of Tyr223 in SMS2. In the SMS2 isoform, Phe224 is responsible for anchoring the phosphocholine head.

The immobilization of the phosphocholine head allows the release of DAG and the access of Cer in the catalytic site.

The two isoforms show no obvious differences in the type and duration of the interactions they establish with the PC. In both cases, the presence of tyrosine is pivotal to maintain continuous contact with the phosphocholine head (Figure 4b,c). These interactions last for more than 90.0% of the simulation time.

### 3.4. Free Energy Profile: Hydroxylated Ceramide Is the Better Substrate of SMSs

All molecules involved in the conversion of PC into SM have been docked to the receptors. The fatty acid (FA) chains chosen were palmitic and oleic, representing the two most common FA found in human plasma membranes. The estimation of the binding energy, although not accurate in absolute value, allows us to reconstruct the energy profile of the reaction. The structures obtained were minimized, and the binding energy of each molecule was calculated (for more details on the systems’ building, see the section “Molecular docking” in the Materials and Methods section).

The SMS enzyme activities are schematized in Figure 5. PC is the natural substrate of SMS, and it can move freely in the membrane (1) to reach the enzyme’s catalytic site (2). The phospholipid head is transferred from the PC to a tyrosine (Tyr223 for SMS1 and Tyr167 for SMS2), leading to the formation of DAG and phosphorylated SMS enzyme (indicated as SMS-P) (3). Then, the DAG leaves the binding site heading into the membrane (4). The second substrate is also a component of cell membranes. Ceramide (Cer) moves from the membrane (5) to the catalytic site of the activated form of the enzyme (6). The phosphocholine head is transferred to ceramide to form SM (7). Restoration of the unphosphorylated form of SMS is completed with SM moving into the membrane (8).

To study the energy pattern of this reaction, we analyzed three different systems. The first system represents the non-hydroxylated substrates (Figure 6a). In detail, the box in Figure 6a shows the sum of the binding energy of PC and Cer in the membrane (PC + Cer) m (ΔG = −21.7 kcal/mol) and the sum of the energy of binding of DAG and SM in membrane (DAG + SM)m (ΔG = −26.6 kcal/mol). The m as a subscript in parentheses indicates substrates immersed in a POPC membrane. The second system hypothesizes the incorporation of 2-hydroxy oleic acid (2OHOA) into the PC substrate, at C2 in the R configuration (2ROHPC) and in the S configuration (2SOHPC) (Figure 6b,c). The presence of hydroxyl group at C2 of PC leads to hydroxylated DAG (2ROHDAG and 2SOHDAG, respectively) and non-hydroxylated SM. In the third system, we started with hydroxylated ceramides (Figure 6d,e). Hydroxylated ceramides produce two hydroxylated SM (in R and S configuration). The values of ΔΔG are calculated as the difference between the binding energy of the DAG, SM pair, and the PC, Cer pair immersed in a POPC membrane.

We observed that for non-hydroxylated substrates, the equilibrium of the reaction is shifted toward DAG and SM with an energy difference ΔΔG = −4.9 kcal mol^−1^. Hydroxylation of PCs leads to less stable systems. The presence of hydroxyl in ceramide shifts the balance toward products by approximately 0.6–0.9 kcal mol^−1^. The stereochemistry of hydroxylation has only a negligible effect. 2OHOA administration has been observed to increase SM levels in the cell [62,63]. Based on our findings, the increase in SM levels is consistent with the incorporation of 2OHOA into ceramide. This suggests a possible role for ceramide synthase that will be investigated in an upcoming paper.

The differences between the two SMS isoforms can be similarly investigated by calculating the binding energies in steps 2, 3, 6, and 7, as shown in Figure 5 for non-hydroxylated and hydroxylated lipids on C2. In detail, we compared the binding energies of PC and SM in complex with SMS (SMSx/PC and SMSx/SM, respectively) and the binding energy of the phosphorylated isoform (SMSx-P) in complex with Cer and DAG (SMSx—P/Cer and SMSx—P/DAG, respectively) (Figure 7).

The SMS-catalyzed reaction leading to the formation of SM and DAG involves several intermediate states. First, the PC must bind to SMS (SMSx/PC), lose the phosphocholine head with the formation of a covalent bond (SMSx—P/DAG), and depart transformed into DAG. A second non-covalent binding event is then required in which Cer enters the active site (SMSx—P/Cer), acquires the choline group that was bound to the enzyme, and departs as SM (SMSx/SM). The covalent binding between SMS and choline should not be counted because, overall, it does not contribute to the change in energy of the system. Figure 7 shows the difference in binding efficiency between products and reactants in the receptor. The equilibrium energy values indicated an energy preference of the interaction of the products with the receptor and were used to derive the kinetic constants used in the section. Interesting was that the potential production of a hydroxylated SM is energetically favored, especially for isoform 1 (with a ΔΔG value of −4.0 kcal mol^−1^ for the stereoisomer R and −4.2 kcal mol^−1^ for the stereoisomer S). The energetically favored interaction of hydroxylated ceramides prompted us to investigate the role of hydroxylated ceramides in the transfer of the phosphocholine group from SMSx-P to ceramide.

We used the phosphorylated form of SMS with choline groups covalently bonded to Tyr223 and Tyr167, respectively (indicated as SMS1—P and SMS2—P). The enzymes are fully embedded in a POPC membrane. The binding free energy (ΔG) of the molecules in complex with SMS was calculated using metadynamics.

Metadynamics allows a reconstruction of the free energy profile as a function of two collective variables that describe the movement of the ceramide (and hydroxylated analogs) in the binding site of the phosphorylated protein. We chose as collective variables (CVs) two distances. CV1 is the distance between the phosphorous of the modified tyrosine residue (Tyr223 for SMS1 and Tyr167 for SMS2) and the oxygen atom of the ceramide (atom O5), and CV2 is the distance between the oxygen atom in the P=O group of the modified tyrosine and the oxygen atom of the hydroxyl group of the sphingosine chain (see Appendix A).

The minima location observed for non-hydroxylated ceramide confirms the mechanism proposed in SMS1 and SMS2 (Figure 8). The isoform SMS1-P is the enzyme with the best binding affinity for ceramide (of −9.33 kcal/mol, Figure 8a), showing two energy minima in the proximity of the key residue identified in this work. On the other hand, isoform SMS2—P has a higher binding affinity for the hydroxylated form of the natural substrate (of −9.52 kcal/mol, Figure 8e).

The metadynamics results have two important consequences. First, the presence of a hydroxyl group allows effective binding (i.e., at a position and distance useful for choline group transfer) to SMS2 (see Figure 8e,f). Indeed, the presence of a hydroxyl group in the R configuration in ceramide leads to the formation of a hydrogen bond with the carbonyl backbone of the isoleucine Ile 207. This interaction frees up space for a water molecule to replace the OH group of the ceramide in the tyrosine 167 binding to the choline head (see Appendix A). The activation of SMS2, which is primarily located in the plasma membrane, results in an increase in the rate of PC→SM conversion and a modulating effect on the physical state of the plasma membrane. It is well known that the composition of the plasma membrane plays a key role in controlling the functionality of numerous proteins [6,7,64].

Second, the effective binding of hydroxylated ceramide on SMS2 can be modeled by an increase in the rate constant of the reaction that transforms Cer into SM. The effect on sphingolipid levels is investigated below using ordinary differential equations.

### 3.5. The SMS Pathway

The results described above suggest an increase in the reaction rate by which SM yields the choline group to the enzyme. To evaluate the effects of increasing the kinetic constant, we conducted a pharmacodynamic analysis. Reaction mechanisms have been studied for several decades. However, in drug discovery, they are often surprisingly neglected when interpreting the results of molecular docking or molecular dynamics. Since Copeland’s work on residence time [65], the kinetic aspects of ligand–receptor interaction have gained some popularity. Still, system chemistry has no place in the design of new drugs. This leads many researchers to use models of action inadequate and not supported by a satisfactory statistical analysis.

In the SMS example, the overall reaction is the following:(1)$$PC+Cer\;\begin{array}{c} \overset{k_{on}}{\to }\\ \underset{k_{off}}{\leftarrow } \end{array}DAG+SM$$

That should be modeled in (at least) two stages as follows:(2)$$PC+SMS\;\begin{array}{c} \overset{k_{1\;}}{\to }\\ \underset{k_{-1\;}}{\leftarrow } \end{array}SMSP+DAG$$
and
(3)$$Cer+SMSP\;\begin{array}{c} \overset{k_{2}}{\to }\\ \underset{k_{-2}}{\leftarrow } \end{array}SMS+SM$$

The equilibrium constant is calculated as the ratio of concentrations of products to reactants, but this is seldom achieved because the system is continuously fed from outside. The rate constants for all intermediate stages can be estimated by calculations. Their importance is inversely proportional to their magnitude because the rate of a series of processes cannot be greater than the slowest step. A ligand capable of modifying the kinetics constant of the slow step can significantly affect the concentrations of the species present. As we have seen through metadynamics, a hydroxyl group in 2ROHCer suggests an increase of k_on_. Though the actual value cannot be accurately determined, we can test the hypothesis that 2ROHCer might alter the levels of SM and DAG, Cer, and PC.

In the example shown in this work, we considered an equilibrium constant K = 0.35 for the overall reaction of Equation (1). The value is a reasonable guess based on literature analysis [63,66,67], and it simply serves as a check on the functioning of the network described below.

The velocity coefficients are initially set to k_1_ = 0.1, k_−1_ = 0.2, k_2_ = 1, k_−2_ = 1.43. These values, obtained from an estimate of the energy of the activated state, are consistent with the equilibrium constant of the reaction in Equation (1). The overall k_on_ and k_off_ constants are related to these by the relations:(4)$$k_{on}=min\left\{\;k_{1},k_{2}\ldots .\;k_{n}\;\right\}$$
(5)$$k_{off}=min\left\{\;k_{-1},k_{-2}\ldots .\;k_{-n}\;\right\}$$

Once the rate constants are fixed, the concentrations at equilibrium are determined. In Figure 9, we started from a percent composition of 25% for the four species, and the concentrations evolve until the relationship expressed by the equilibrium constant is satisfied. To simulate the different binding of 2ROHCer as observed in Figure 8e, we increase the k_2_ to 2. The new concentrations are radically different, and we observe a significant increase in SM and DAG (Figure 9).

As can be clearly seen in Figure 9, the better binding of 2ROHCer, and consequently a higher k_2_ constant, leads to a decrease in PC and Cer levels and an increase in SM and DAG levels. This analysis allows us to clarify the role of 2OHOA in the recovery of lipid homeostasis. Experimental data, although sometimes contradictory, indicate a change in SM and PC levels after 2OHOA addition. Some work [62,63] suggests an increase in SM levels and decreased PC levels. This observation immediately suggested SMS as a potential target of 2OHOA. Docking, per se, offers no indication of the actual interaction of a ligand with a receptor, and merely estimates the geometry and intensity of the interaction. The binding of free 2OHOA is significantly lower than that of, e.g., hydroxylated ceramide. Therefore, there is no indication nor computational nor experimental, that free 2OHOA can exist in a cell. Instead, theoretical data indicate enzymes that incorporate fatty acids such as ceramide synthase as possible targets of 2OHOA. The interaction of SMS with the most common sphingolipids was evaluated. As already shown, hydroxylated ceramide is allocated in a position not suitable for choline head recovery. A higher k_2_ rate constant leads to increased SM and DAG levels in plasma membrane. In this sense, the 2OHOA behaves as if it was an activator of SMS. These findings are also in line with the proposed working model of Ou et al. [68] that suggests the requirement of DAG on the plasma membrane for activation of protein kinase C δ (PKCδ) nuclear translocation.

## 4. Conclusions

The 3D structures of SMS1 and SMS2 have been refined and used for computational investigation, and the binding site of both SMS1 and SMS2 was found in the transmembrane region. We have found a key tyrosine involved in the SMS mechanism (Tyr223 for SMS1 and Tyr167 for SMS2). This was demonstrated by molecular dynamics that also showed a water molecule in the catalytic area, assisting the nucleophilic attack on the phosphocholine.

We determined the energy profile of the PC→SM transformation and defined the intermediate steps in both SMS1 and SMS2.

An interesting difference between SMS1 and SMS2 towards the hydroxylated species appears from the metadynamics results. We showed that hydroxylated ceramide, especially in the R configuration, is largely favored in the interaction with SMS2. As SMS2 is mainly localized in the plasma membrane, our results suggest a possible role of some hydroxylated species in the homeostasis of lipid composition. In addition, the role of 2OHOA can also be investigated with respect to its incorporation into ceramide.

In this work, structural, thermodynamic, kinetic, and system aspects have been combined to provide a comprehensive view of the action of the SMS and the possible role of 2OHOA. Much more can be done by extending the network to other receptors and lipid species. Advances in lipidomics and computational calculations make it possible to simulate the system-wide effect of ligand–receptor binding in real-time. Conversely, the presence of experimental data allows for identifying new potential targets for diseases related to lipid metabolism. Data in the literature confirm the production of hydroxylated R-acyls by the stereospecific enzyme FA2H. Our results suggest a possible role for another enzyme, ceramide synthase, in incorporating 2OHOA (particularly in the R form) into ceramide.

These findings pave the way for a better understanding of the role of 2OHOA and, more generally, hydroxylated sphingolipids in the mechanisms controlling autoimmunity in healthy individuals.

## Figures and Tables

**Figure 1 membranes-11-00787-f001:**
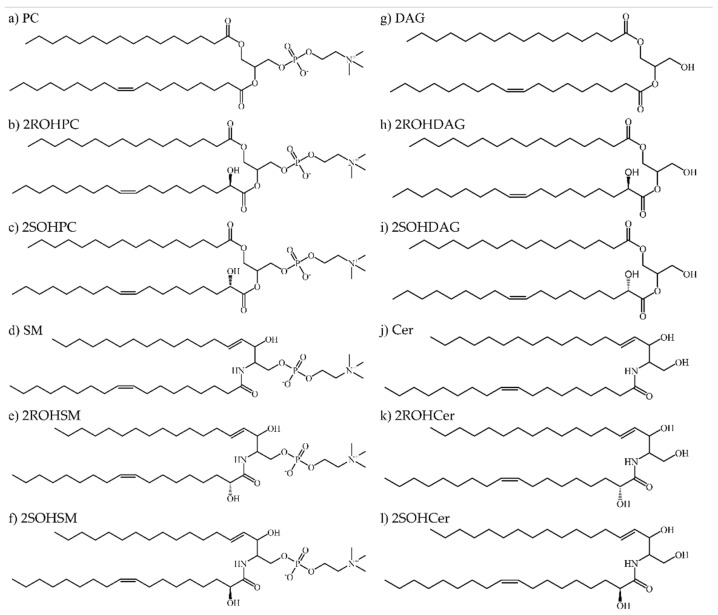
Chemical structures of the ligand set. (**a**) 1-palmitoyl-2-oleoyl-sn-glycero-3-phosphocholine (PC), (**b**) hydroxylated PC with 2ROHOA (2ROHPC), (**c**) hydroxylated PC with 2SOHOA (2SOHPC), (**d**) *N*-oleoyl-d-erythro-sphingosylphosphorylcholine (SM), (**e**) hydroxylated SM with 2ROHOA (2ROHSM), (**f**) hydroxylated SM with 2SOHOA (2SOHSM). (**g**) (S)-1-hydroxy-3-(palmitoyloxy)propan-2-yl oleate (DAG), (**h**) hydroxylated DAG with 2ROHOA (2ROHDAG), (**i**) hydroxylated DAG with 2SOHOA (2SOHDAG), (**j**) ceramide *N*-((2S,3R,E)-1,3-dihydroxyoctadec-4-en-2-yl)oleamide (Cer), (**k**) hydroxylated Cer with 2ROHOA (2ROHCer), (**l**) hydroxylated Cer with 2SOHOA (2SOHCer).

**Figure 2 membranes-11-00787-f002:**
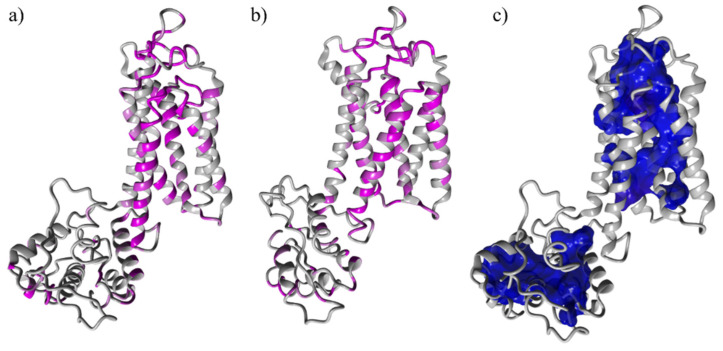
Conserved residues represented in magenta for SMS1 (**a**) and SMS2 (**b**). In blue, the surface of SMS1 protein residues close to water molecules with low mobility (**c**).

**Figure 3 membranes-11-00787-f003:**
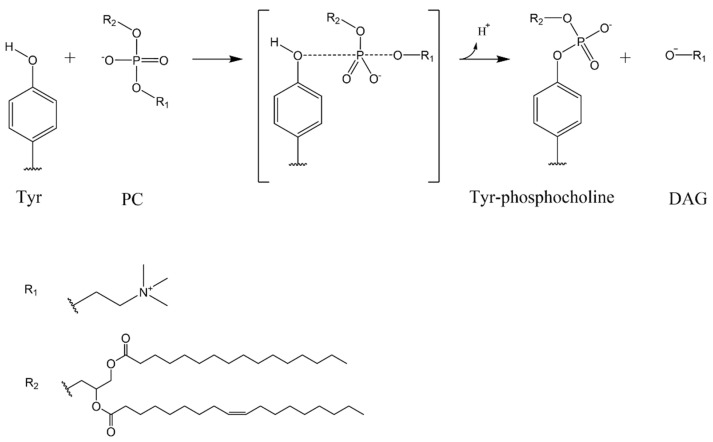
The hypothesis of the first step of the mechanism for SMS enzyme. The reaction starts with a nucleophilic attack of the hydroxyl group of a tyrosine in the binding site to the phosphorus atom of a PC molecule. The result is the formation of DAG and phosphorylation of the enzyme.

**Figure 4 membranes-11-00787-f004:**
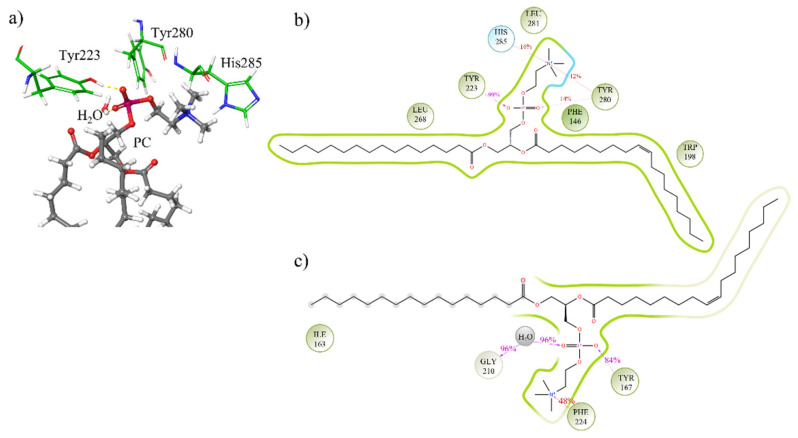
A closeup of the interactions between PC and key residues at the binding site in SMS1 (**a**). Two-dimensional interaction map of ligand-protein contacts for SMS1/PC (**b**) and SMS2/PC (**c**).

**Figure 5 membranes-11-00787-f005:**
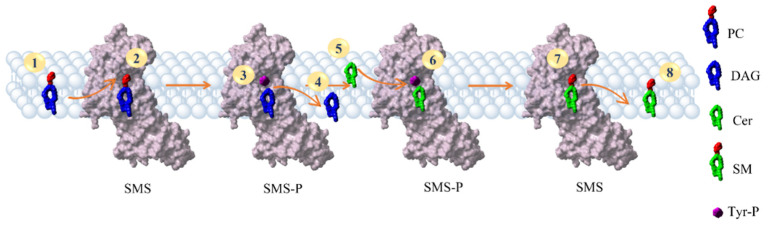
Overview of the SMS reaction steps. In step (1), one of the PC molecules was highlighted. In step (2), the PC at the binding site of the SMS enzyme is observed. The transfer of the phosphocholine head to the tyrosine of the enzyme forms a modified tyrosine shown in magenta in step 3. The removal of the phosphocholine head from the PC generates DAG (3). Thus, it is free to move through the membrane (4). In steps 5 and 6, ceramide moves from the membrane (5) to the SMS (6). The transfer of the phosphocholine head from the modified tyrosine to Cer generates SM (7), followed by its release into the membrane (8).

**Figure 6 membranes-11-00787-f006:**
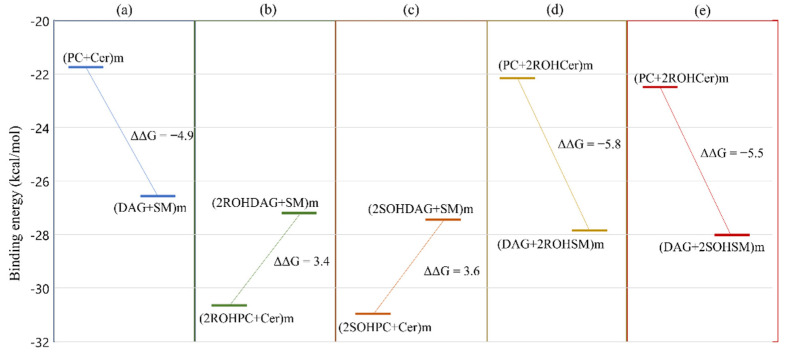
Free energy differences for different hydroxylation paths in POPC membrane: non-hydroxylated system (**a**); hydroxylated PC with R configuration (**b**) and with S configuration (**c**); hydroxylated ceramide with R configuration (**d**) and S configuration (**e**). For the chemical structures of the lipid, see Figure 1.

**Figure 7 membranes-11-00787-f007:**
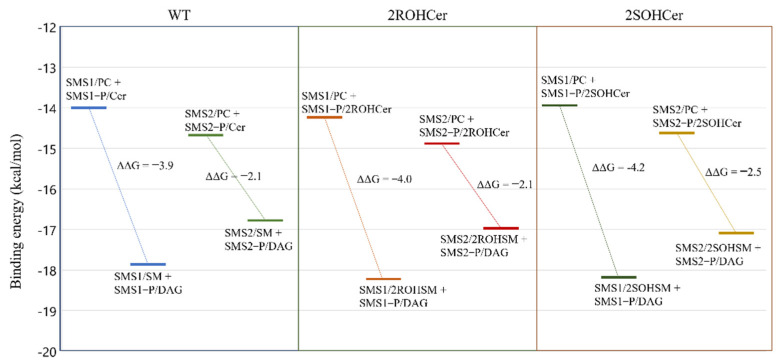
Free energy of intermediate steps involving the SMS isoforms. On the left (WT) is the non-hydroxylated system; in the middle and on the right, the ceramide hydroxylated in position 2 with R configuration (2ROHCer), and S configuration (2SOHCer), respectively.

**Figure 8 membranes-11-00787-f008:**
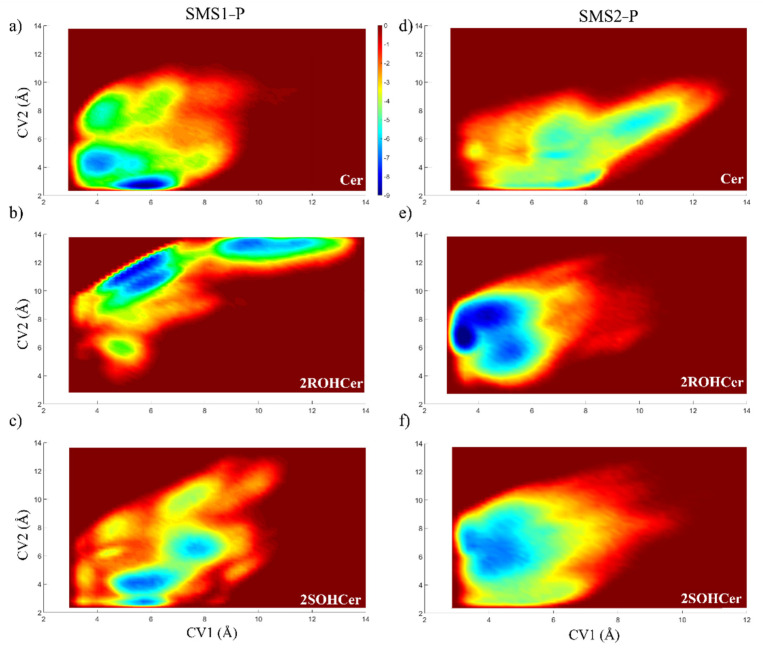
Two-dimensional free-energy surface of SMS1 in complex with (**a**) ceramide, (**b**) ceramide 2R hydroxylated, (**c**) ceramide 2S hydroxylated; SMS2 in complex with (**d**) ceramide, (**e**) ceramide 2R hydroxylated, (**f**) ceramide 2S hydroxylated. The CV1 and CV2 are the *x*- and *y*-axis, respectively.

**Figure 9 membranes-11-00787-f009:**
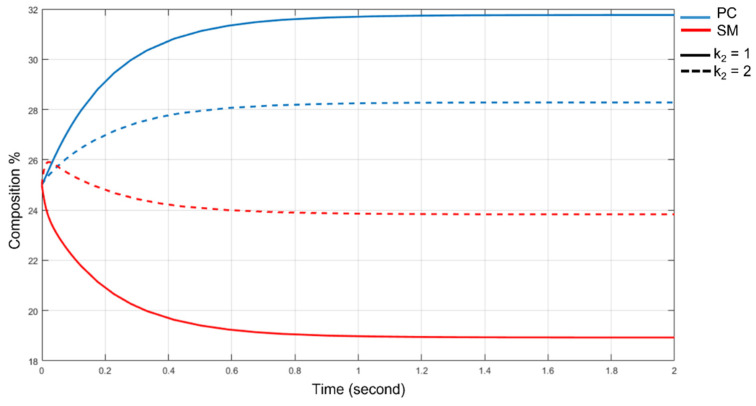
Standard output of SimBiology tool to represent the SMS reaction trend in normal condition (solid line) and for higher k_2_ (dashed line). On the *x*-axis, the time in seconds, on the *y*-axis, the species amounts expressed as % mol/mol. The colors of the lines have been assigned as follows: red for SM, and blue for PC.

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
