# Peer review of "Hydroxylated Fatty Acids: The Role of the Sphingomyelin Synthase and the Origin of Selectivity"

_membranes, 2021, doi:10.3390/membranes11100787_

Round 1

Reviewer 1 Report

The research article by L. Sessa et al. describes a detailed study about the role of hydroxylated lipids in sphingomyelin synthase. In particular, the mechanism behind the interaction between hydroxylated lipids and two isoforms of SMS enzymes, SMS1 and SMS2 has been illustrated. The authors also provide the structure prediction of the two isoforms as well as the binding site identification and the specific role of Tyrosine residue within the mechanism of action. In general, the manuscript is well-written, well-organized and properly referenced even if it can be slightly improved in the introductive part. The results allow to shed light on the role of hydroxylation on the alpha carbon in all the structural components involved, namely sphingomyelin, phosphatidylcholine and ceramide; the discussions are well-founded and the conclusions are supported by results. Nevertheless, in some points the discussion can be slightly broadened, e.g.expanding references to highlight the importance and the relative complexity of sphingomyelin within biological membranes. Recommendations to improve the quality of the paper are listed below:

-Introduction section, lines 32-35: authors refer to the crucial role of sphingolipids in determining the membrane properties and they introduce the specific case of sphingomyelins which increase the stiffness and the compactness of membrane lipids. For completeness, authors could refer also to almost recent findings about a very complex behavior of GUV containing different kind of sphingomyelins which leads, at specific combinations of composition and temperature, to three-phase coexistence regions as well as to the role played by acyl chain mismatch on the formation of two-dimensional domains. You may critically use following papers:(Biophysical Journal 116, 3, 503-517 - https://doi.org/10.1016/j.bpj.2018.12.018; Biophysical Journal 119, 5, 913-923 - https://doi.org/10.1016/j.bpj.2020.07.014). A brief mention to these or similar cases, would help not-in-the-field readers in pointing out the complexity and the importance of sphingolipids and in particular of sphingomyelins.

-Figure 4: in panels b) and c) the graphical resolution is too low and does not allow to read inside the colored spheres. It would be nice to get the message even at graphical level; if it is tough to modify I suggest to add a legend.

-Figure 8: to facilitate the reading, labels indicating cer, cer2R and cer2S should be added directly on the figure panels.

-Figure 9: At time close to zero the reaction trend for K2 reveals an initial increase; may the authors explain this behavior? Is it relevant? Why it does not happen for K1?

Author Response

Answers to Reviewer 1

We wish to thank the reviewer for his thoughtful comments on the original version of our paper entitled Hydroxylated fatty acids: the role of the sphingomyelin syn-2 thase and the origin of selectivity.

We carefully considered all comments and suggestions of the reviewer. Herein, we explain how we revised the paper based on these comments and recommendations. As a result, we believe that the manuscript edited in line with all Reviewers' comments has considerably improved and now reaches the bar of acceptance.

POINT-BY-POINT RESPONSES TO REVIEWER 1

 Nevertheless, in some points the discussion can be slightly broadened, e.g.expanding references to highlight the importance and the relative complexity of sphingomyelin within biological membranes. Recommendations to improve the quality of the paper are listed below:

-Introduction section, lines 32-35: authors refer to the crucial role of sphingolipids in determining the membrane properties and they introduce the specific case of sphingomyelins which increase the stiffness and the compactness of membrane lipids. For completeness, authors could refer also to almost recent findings about a very complex behavior of GUV containing different kind of sphingomyelins which leads, at specific combinations of composition and temperature, to three-phase coexistence regions as well as to the role played by acyl chain mismatch on the formation of two-dimensional domains. You may critically use following papers:(Biophysical Journal 116, 3, 503-517 - https://doi.org/10.1016/j.bpj.2018.12.018; Biophysical Journal 119, 5, 913-923 - https://doi.org/10.1016/j.bpj.2020.07.014). A brief mention to these or similar cases, would help not-in-the-field readers in pointing out the complexity and the importance of sphingolipids and in particular of sphingomyelins.

As suggested by the reviewer, we have included new references, including the two references above, to mention the role of sphingomyelin in modulating the physical state of membranes.

-Figure 4: in panels b) and c) the graphical resolution is too low and does not allow to read inside the colored spheres. It would be nice to get the message even at graphical level; if it is tough to modify I suggest to add a legend.

We modified Figure 4, as suggested by the reviewer.

-Figure 8: to facilitate the reading, labels indicating cer, cer2R and cer2S should be added directly on the figure panels.

We modified Figure 8, as suggested by the reviewer.

-Figure 9: At time close to zero the reaction trend for K2 reveals an initial increase; may the authors explain this behavior? Is it relevant? Why it does not happen for K1?

Solving systems of differential equations often leads to transient and nonlinear behavior. However, these trends are of little relevance to our discussion, which focuses on how the interaction of certain ligands with the receptor manages to alter the concentration values at equilibrium.

Reviewer 2 Report

The authors present a computational study of SMS1 and SMS2 towards understanding the mechanism of their enzyme-catalyzed reactions and determining binding energies of common ligands. The computational methods employed by the authors appear rigorous and thorough. However, major improvements in the presentation of the results are needed. Also, there are numerous statements in the introduction that require corresponding references. In addition, a large number of self citations occurred: 13 self-citations, representing about 23% of the references. 

Major revisions: 

1) Appropriate references need to be added to back up several statements in the introduction. See the sentences ending on line 30, line 37, line 52, line 55, line 74, and line 82. 

2) The authors state on line 236-238 that they previously obtained a structure of SMS1 through homology modeling but the new model using deep-learning if much more accurate. How do you know the structure is more accurate? How different is the new model to the previous homology model? An RMSD of the alignment between the two would be helpful here. 

3) Asn265 is noted in the text on line 309 and Figure 4a is referenced, however Asn265 is not displayed in the figure. Rather His 285 is shown but not discussed anywhere in the text. 

4) The binding energies that were calculated from the molecular docking are not presented anywhere. This data would help demonstrate how 2OHOA binding energy compares to other ligands. You say in line 477-478 that "evidence does not support the claim that 2OHOA can interact directly with SMS". But you docked this ligand to SMS, so wouldn't that provide evidence of interaction? 

3) Figure 9 shows an increase in concentration of PC and decrease of SM over time. But SMS catalyzes the conversion of PC (substrate) to SM (product) based on your schematic in Figure 5 and equilibrium equation. So if this Figure is meant to show the output of an SMS reaction how its is possible that the % composition of reactant is increasing and the product is decreasing?

4) There are a large number of self-citations that occur in the references of this paper. A total of 13 were counted out of 57 (~22% of the references) 

Minor revisions:

1) A figure showing chemical structures of the various lipids studies in the paper that corresponds with the discussion of the various types of lipids and locations of hydroxylation on lines 67-69 would be very helpful. 

2) In line 81 you introduce the acronym FA2H without first defining it. 

3) Figure 1 does not significantly add to the manuscript and should be moved to the SI. Replace with chemical structures of lipids with appropriate labels and acronyms. 

4) On line 291 the authors say: "Autodock VINA represents the golden standard in molecular docking". This seems like a very bold statement to make without having information to back this up. 

4) The authors specifically note a water molecule near Tyr 223 in line 302 but do not explain the exact chemical role it is playing the reaction or show it in Figure 3. 

5) The labels of the amino acids on the LigPlots shown in Fig 4b, c and S4 are way too small and impossible to read. These figures need to be redone so they are readable. 

6) Figure 6 and 7 could use more detailed explanations of the various acronyms displayed in the figure legends. Also, what does the m mean outside the parenthesis in Figure 6?

7) In line 418 you say there is a hydrogen bond with an isoleucine. What is the residue number? I assume this is with the main chain of the amino acid? If so, then you should say with the carbonyl or NH backbone. 

8) In line 460, you say you start with concentrations of 25%. Percentages are not the same as concentrations, change this to "a percent composition of 25% for each of the 4 species"

Author Response

Answers to Reviewer 2

We wish to thank the Reviewer for his thoughtful comments on the original version of our paper entitled Hydroxylated fatty acids: the role of the sphingomyelin synthase and the origin of selectivity.

We carefully considered all comments and suggestions of the Reviewer. Herein, we explain how we revised the paper based on these comments and recommendations. As a result, we believe that the manuscript edited in line with all Reviewers' comments has considerably improved and now reaches the bar of acceptance.

POINT-BY-POINT RESPONSES TO REVIEWER 2

Major revisions: 

1) Appropriate references need to be added to back up several statements in the introduction. See the sentences ending on line 30, line 37, line 52, line 55, line 74, and line 82. 

As suggested by the Reviewer, we have included 10 new references at the indicated positions.

2) The authors state on line 236-238 that they previously obtained a structure of SMS1 through homology modeling but the new model using deep-learning if much more accurate. How do you know the structure is more accurate? How different is the new model to the previous homology model? An RMSD of the alignment between the two would be helpful here. 

The structures used in the manuscript were obtained with Folden software, which uses neural networks to evaluate correspondences and orientations between residuals. Finally, we compared the two structures using the Whatif validation tool. Below is the output.

Actual structure

Previous structure

Structure Z-scores, positive is better than average:

  1st generation packing quality :  -1.538

  2nd generation packing quality :  -1.675

  Ramachandran plot appearance   :  -0.541

  chi-1/chi-2 rotamer normality  :   1.311

  Backbone conformation          :  -1.966

 RMS Z-scores, should be close to 1.0:

  Bond lengths                   :   1.127

  Bond angles                    :   0.442 (tight)

  Omega angle restraints         :   1.469 (loose)

  Side chain planarity           :   0.943

  Improper dihedral distribution :   0.572

  Inside/Outside distribution    :   1.237 (unusual)

Structure Z-scores, positive is better than average:

  1st generation packing quality :  -3.250

  2nd generation packing quality :  -3.718 (poor)

  Ramachandran plot appearance   :  -3.698 (poor)

  chi-1/chi-2 rotamer normality  :  -3.618 (poor)

  Backbone conformation          :  -3.838 (poor)

 RMS Z-scores, should be close to 1.0:

  Bond lengths                   :   1.174

  Bond angles                    :   0.675

  Omega angle restraints         :   1.655 (loose)

  Side chain planarity           :   1.513

  Improper dihedral distribution :   1.156

  Inside/Outside distribution    :   1.276 (unusual)

3) Asn265 is noted in the text on line 309 and Figure 4a is referenced, however Asn265 is not displayed in the figure. Rather His 285 is shown but not discussed anywhere in the text. 

We thank the reviewer very much for pointing out this typo. We have corrected the text by reporting the amino acid His 285, agreeing with Figure 4a and our trajectory analysis.

4) The binding energies that were calculated from the molecular docking are not presented anywhere. This data would help demonstrate how 2OHOA binding energy compares to other ligands. You say in line 477-478 that "evidence does not support the claim that 2OHOA can interact directly with SMS". But you docked this ligand to SMS, so wouldn't that provide evidence of interaction? 

Docking, per se, offers no indication of the actual interaction of a ligand with a receptor and merely estimates the geometry and intensity of the interaction (in kcal/mol). The binding of free 2OHOA is significantly lower than that of, e.g., hydroxylated ceramide. More important to note is that there is no indication that free 2OHOA can exist in a cell, but it is readily incorporated into membrane lipids. Therefore, the observed in vitro effect of 2OHOA is presumably a consequence of its incorporation into Ceramide and subsequent interaction with (at least) SMS. We have changed the sentence in agreement.

3) Figure 9 shows an increase in concentration of PC and decrease of SM over time. But SMS catalyzes the conversion of PC (substrate) to SM (product) based on your schematic in Figure 5 and equilibrium equation. So if this Figure is meant to show the output of an SMS reaction how its is possible that the % composition of reactant is increasing and the product is decreasing?

Inhibition, even partial, of SMS, leads to an accumulation of reactants and a reduction of products. Thus, SMS, like any enzyme, offers a pathway for the transformation of reagents and products with lower activation energy. This pathway can also be used for the transformation of products into reagents. The speed of the two processes is expressed by the two kinetic constants kon and koff. If either of these constants is altered, the equilibrium of the reaction (which is the ratio of the constants) is also altered.

4) There are a large number of self-citations that occur in the references of this paper. A total of 13 were counted out of 57 (~22% of the references) 

Our research on the physical state of membranes and how this affects the activity of proteins began 16 years ago and has covered many aspects of research relevant to this work, such as the definition of the physical state of membranes with Laszlo Vigh and coworkers, the role of 2OHOA with Pablo Escribà and coworkers, molecular dynamics and stochastic simulations with Fabio Mavelli, up to the definition of new docking methods and new software. Each reference inserted has a precise meaning in the economy of the manuscript.

Minor revisions:

1) A figure showing chemical structures of the various lipids studies in the paper that corresponds with the discussion of the various types of lipids and locations of hydroxylation on lines 67-69 would be very helpful. 

According to the Reviewer's suggestion, we included the structures of the studied lipids in Figure 1.

2) In line 81 you introduce the acronym FA2H without first defining it. 

According to the Reviewer's suggestion, we added the acronym of FA2H in the text.

3) Figure 1 does not significantly add to the manuscript and should be moved to the SI. Replace with chemical structures of lipids with appropriate labels and acronyms. 

We accepted the reviewer's suggestion and moved Figure 1 to the supplementary materials. We have renumbered the SI images accordingly. The chemical structures of the ligands have been included in the new Figure 1, in section 2.3.

4) On line 291 the authors say: "Autodock VINA represents the golden standard in molecular docking". This seems like a very bold statement to make without having information to back this up. 

We agree with the reviewer and have rephrased the sentence.

4) The authors specifically note a water molecule near Tyr 223 in line 302 but do not explain the exact chemical role it is playing the reaction or show it in Figure 3. 

We added some words to explain this point better.

5) The labels of the amino acids on the LigPlots shown in Fig 4b, c and S4 are way too small and impossible to read. These figures need to be redone, so they are readable. 

We modified Figure 4, as suggested by the Reviewer.

6) Figure 6 and 7 could use more detailed explanations of the various acronyms displayed in the figure legends. Also, what does the m mean outside the parenthesis in Figure 6?

The m as a subscript in parentheses indicates substrates immersed in a POPC membrane. We have rewritten the text to better describe Figures 6 and 7.

  7) In line 418 you say there is a hydrogen bond with an isoleucine. What is the residue number? I assume this is with the main chain of the amino acid? If so, then you should say with the carbonyl or NH backbone. 

We rewrote the sentence by inserting the residue number Ile207 and defining the type of carbonyl bonding of the residue backbone. We have improved the resolution of Figure S5.

8) In line 460, you say you start with concentrations of 25%. Percentages are not the same as concentrations, change this to "a percent composition of 25% for each of the 4 species"

We agree with the reviewer and have corrected the text accordingly.

Round 2

Reviewer 2 Report

The authors have significantly improved the manuscript with their revisions and adequately addressed all points raised with one exception, listed below. 

1) Include a description of the WhatIf analysis of the SMS1 structures in the methods section and include the output of results in the Supplemental material. Include information regarding the alignment of the previous structure and new structure (RMSD value or something similar). 

Author Response

Following the reviewer's suggestion we inserted a sentence in line 111 to introduce WHAT IF and we added the output as Table S1.

We renumbered all Table in the manuscript accordingly.